# Design of Hierarchical NiCo_2_O_4_ Nanocages with Excellent Electrocatalytic Dynamic for Enhanced Methanol Oxidation

**DOI:** 10.3390/nano11102667

**Published:** 2021-10-11

**Authors:** Xue Li, Gege He, Chong Zeng, Dengmei Zhou, Jing Xiang, Wenbo Chen, Liangliang Tian, Wenyao Yang, Zhengfu Cheng, Jing Song

**Affiliations:** 1School of Electronic Information and Electrical, Chongqing University of Arts and Sciences, Chongqing 400000, China; s190601011@stu.caupt.edu.cn (X.L.); 20200018@cqwu.edu.cn (C.Z.); 20190012@cqwu.edu.cn (D.Z.); 20190017@cqwu.edu.cn (J.X.); 20160010@cqwu.edu.cn (W.C.); 2School of Science, Chongqing University of Posts and Telecommunications, Chongqing 400065, China; 3School of Physics, Xi’an Jiaotong University, Shanxi 710000, China; gghe01@stu.xjtu.edu.cn; 4Institute of Process Engineering, Chinese Academy of Sciences, Beijing 100190, China; jsong@ipe.ac.cn

**Keywords:** hierarchical hollow nanocages, NiCo_2_O_4_, coordinated etching and precipitation, methanol oxidation, fuel cell

## Abstract

Although sheet-like materials have good electrochemical properties, they still suffer from agglomeration problems during the electrocatalytic process. Integrating two-dimensional building blocks into a hollow cage-like structure is considered as an effective way to prevent agglomeration. In this work, the hierarchical NiCo_2_O_4_ nanocages were successfully synthesized via coordinated etching and precipitation method combined with a post-annealing process. The nanocages are constructed through the interaction of two-dimensional NiCo_2_O_4_ nanosheets, forming a three-dimensional hollow hierarchical architecture. The three-dimensional supporting cavity effectively prevents the aggregation of NiCo_2_O_4_ nanosheets and the hollow porous feature provides amounts of channels for mass transport and electron transfer. As an electrocatalytic electrode for methanol, the NiCo_2_O_4_ nanocages-modified glassy carbon electrode exhibits a lower overpotential of 0.29 V than those of NiO nanocages (0.38 V) and Co_3_O_4_ nanocages (0.34 V) modified glassy carbon electrodes. The low overpotential is attributed to the prominent electrocatalytic dynamic issued from the three-dimensional hollow porous architecture and two-dimensional hierarchical feature of NiCo_2_O_4_ building blocks. Furthermore, the hollow porous structure provides sufficient interspace for accommodation of structural strain and volume change, leading to improved cycling stability. The NiCo_2_O_4_ nanocages-modified glassy carbon electrode still maintains 80% of its original value after 1000 consecutive cycles. The results demonstrate that the NiCo_2_O_4_ nanocages could have potential applications in the field of direct methanol fuel cells due to the synergy between two-dimensional hierarchical feature and three-dimensional hollow structure.

## 1. Introduction

The ever-worsening energy and global warming issues have triggered significant research efforts in the design and development of advanced energy devices. Direct methanol fuel cells (DMFCs) have exhibited great commercialization potential credited to high energy density, low cost, easy storage and low pollutant emissions [1,2]. Generally, the performance of DMFCs was mainly related to the activity of methanol electrocatalyst [3]. Traditionally, platinum group precious metals (Pt, Ru, and Pd, etc.) were always employed as electrocatalysts for methanol. Although high electrocatalytic activity was achieved, the precious metals still suffer from high cost and low working stability [4,5,6]. In this regard, the design of Pt-free catalysts was considered as the best alternative to solve the problems.

Transition metal oxides (TMOs) were recognized as ideal substitutions for noble metals due to their high-active redox sites, low cost and high physicochemical stability. Over the last decade, significant efforts on TMOs have been made to obtain high-performance Pt-free electrocatalysts for methanol [7]. Generally, the nanomaterials in conventional forms of aggregated particles generally have no significant advantages in electrocatalysis. Inspired by kinetics, quantities of TMOs with different microstructures were constructed to improve electrocatalytic kinetics and high electrocatalytic activity was obtained. Thereinto, two-dimensional (2D) nanosheets were demonstrated as ideal structure in electrocatalysis due to the unique physicochemical properties issued from high structural and morphologic anisotropies [8]. However, agglomeration of 2D nanosheets was easy to occur in electrocatalytic reactions because of the large lateral specific surface areas, leading to the decrease of active sites and diffusion channels. 

Integrating amounts of 2D nanosheets into three-dimensional (3D) hierarchical nanocages provided an efficient way to obtain highly active structures. The hierarchical nanocages effectively prevented the aggregation of 2D building blocks and afforded large specific surface areas, which provided sufficient active sites for electrooxidation of methanol [9]. Meanwhile, the pores formed by the interaction of nanosheets not only provided diffusion channels for methanol and intermediate products, but also relieved the volume change and structural strain during electrocatalysis, resulting in excellent stability [10]. Further, the 2D feature of building blocks and porous thin shell of hierarchical nanocages accelerated both the collected and transfer efficiency of catalytic electrons during electrocatalysis, leading to high electrocatalytic activity. Therefore, highly active and stable methanol electrocatalysts can be acquired through the design of hierarchical porous hollow nanocages. 

NiCo_2_O_4_ possesses bimetallic active sites (Co^2+^/Co^3+^ and Ni^2+^/Ni^3+^) and excellent conductivity, exhibiting potential applications in the field of methanol oxidization [11]. In this report, NiCo_2_O_4_ nanocages (NCs) were prepared by coordinated etching and precipitation (CEP) method combined with a post-annealing process. As an electrode for methanol electrooxidation, NiCo_2_O_4_ NCs-modified glassy carbon electrode (GCE) exhibited low overpotential, high current density and excellent stability.

## 2. Materials and Methods

### 2.1. Reagents

NaOH, CuCl_2_·2H_2_O, NiCl_2_·6H_2_O, CoCl_2_·6H_2_O, Na_2_S_2_O_3_·5H_2_O, KOH, methanol and polyvinylpyrrolidone (PVP, *M_w_* = 40,000) were purchased from Chengdu Kelong chemical co. LTD (Chengdu, China) without further purification. L-ascorbic acid (AA) were purchased from Sigma-Aldrich (St. Louis, MO, USA) without further purification.

### 2.2. Preparation of NiCo_2_O_4_ NCs

Cu_2_O templates were firstly prepared according to our previous work [12]. Simply, 10 mL NaOH solution (2 M) was added into 100 mL of CuCl_2_·2H_2_O (0.01 M) and stirred at 55 °C for 30 min. Then, 10 mL of AA (0.6 M) was added. After 3 h of reaction, Cu_2_O cubes were collected and dried in vacuum.

A total of 10 mg cubic Cu_2_O, 1mg NiCl_2_·6H_2_O and 2 mg CoCl_2_·6H_2_O were dispersed into 10 mL ethanol/water (1:1), and then, 0.33 g PVP was added and stirred for 30 min. Afterwards, 4 mL Na_2_S_2_O_3_·5H_2_O solution (1 M) was slowly dropped at room temperature. After 3 h, hydroxide precursors were collected and dried in vacuum. Finally, the precursors were calcined using a tube furnace at 400 °C in air for 2 h with a heating rate of 1 °C min^−1^. Co_3_O_4_ NCs and NiO NCs were respectively prepared as contrast samples using CoCl_2_·6H_2_O and NiCl_2_·6H_2_O only in the CEP process.

### 2.3. Electrochemical Measurements

Cyclic voltammetry (CV), chronoamperometry and electrochemical impedance spectroscopy (EIS) were performed in 1 M KOH solution on CH1760E A191018 electrochemical workstation at room temperature. A three-electrode system was used with Ag/AgCl (saturated with KCl) and platinum disk (*Φ* = 2 mm) as the reference and counter electrodes, respectively. The Co_3_O_4_ NCs, NiO NCs and NiCo_2_O_4_ NCs-modified glassy carbon electrodes (GCE, *Φ* = 3 mm) were applied as working electrodes. Typically, GCE was carefully polished with 3 μm, 0.5 μm and 0.05 μm alumina powders, respectively. Then, 5 μL of the prepared sample suspension (1 mg mL^−1^ in 0.1% Nafion solution) was measured with a pipette and dropped onto the surface of GCE, and then dried naturally. 

### 2.4. Materials Characterization

The microstructures and morphologies of the samples were observed by field emission electron microscope (FESEM, SU8020) and high-resolution transmission electron microscope (HRTEM, FEI F20). The crystal structure and elemental composition were recorded by X-ray powder diffractometer (XRD, Rigaku D/Max-2400 using Cu-Ka radiation λ = 1.54 Å). The chemical state was determined by X-ray photoelectron spectroscopy (XPS, ESCALAB 250Xi) using a 500 μm X-ray spot (energy resolution 0.4 eV). The Brunauer–Emmett–Teller (BET, Belsort-max) was applied to analyze the specific surface area and pore structure.

## 3. Results and Discussion

### 3.1. Characterization

As shown in Figure 1a, Co^2+^ and Ni^2+^ were firstly adsorbed on the surface of Cu_2_O in the ultrasonic process. The CEP process occurred on the surface of Cu_2_O once S_2_O_3_^2−^ was added [13]. Cu_2_O reacts with S_2_O_3_^2−^ and H_2_O to form a soluble [Cu_2_(S_2_O_3_^2−^)_x_]^2−2x^ complex and abundant OH^-^ (reaction (1)). The part-hydrolyzation of S_2_O_3_^2−^ also facilitated the supply of OH^−^ (reaction (2)) [14]. Reactions (1) and (2) concurrently pushed reaction (3) forward, facilitating the formation of Ni-Co hydroxide precursor. The diffusion of S_2_O_3_^2−^ from the surface into the interior of the shell directly affected the rate of Cu_2_O etching, while the transport of OH^−^ from internal to external sites promoted the growth of Ni-Co hydroxide precursor [15]. The two reaction processes coordinated together to achieve dynamic balance to promote the formation of hollow structure. In order to confirm the formation mechanism, the precipitate prepared at 0, 10, 20, 30, and 180 min was collected and observed by TEM (Figure 1b). With the introduction of S_2_O_3_^2−^, Cu_2_O was gradually etched into the polyhedral structure due to the higher diffusion intensity of ions at the corners [15,16]. After the reaction lasted for 3 h, Cu_2_O completely disappeared and hierarchical porous nanocages were obtained (Appendix A). Finally, NiCo_2_O_4_ NCs were obtained through the annealing of Ni-Co hydroxide precursor (reaction (4)). As observed in Figure 1c, the color of the reaction system gradually became shallow and the light green precipitates generated at the same time. The fading was attributed to the etching of Cu_2_O, while the green precipitates were correlated to the formation of Ni-Co hydroxide precursor.
(1)$${\mathrm{Cu}}_{2}{O+S}_{2}O_{3}^{2-}{+H}_{2}O\to {{[\mathrm{Cu}}_{2}{(S}_{2}O_{3}^{2-})x]}^{2-2x}{+2\mathrm{OH}}^{-}$$
(2)$$S_{2}O_{3}^{2-}{+H}_{2}O↔{\mathrm{HS}}_{2}O_{3}^{-}{+\mathrm{OH}}^{-}$$
(3)$${\mathrm{Ni}}^{2+}{+2\mathrm{Co}}^{2+}{+6\mathrm{OH}}^{-}{+1/2O}_{2}\to {(\mathrm{NiCo}}_{2}{)O}_{2}{\left(\mathrm{OH}\right)}_{4}{+H}_{2}O$$
(4)$${(\mathrm{NiCo}}_{2}{)O}_{2}{\left(\mathrm{OH}\right)}_{4}\to {\mathrm{NiCo}}_{2}O_{4}{+2H}_{2}O$$

As shown in Figure 2a, the strong peaks from the templates at 30°, 37°, 42°, 62°, 74° and 78° matched well with PDF#77-0199 of cubic Cu_2_O. As observed in Figure 2b, no significant diffraction peaks were observed in the precursor, revealing poor crystallinity of Ni-Co hydroxide precursor. After calcination, the crystallinity of materials was obviously improved and the diffraction peaks at 36°, 43°, 64°, 75° and 77° were well indexed to the (111), (200), (220), (311) and (400) crystal planes of face-centered cubic NiCo_2_O_4_. XRD results clearly demonstrated the formation of high purity NiCo_2_O_4_ product. Furthermore, XPS measurements were performed to obtain detailed information on the elements and oxidation state of prepared NiCo_2_O_4_. As shown in Figure 2c, the survey spectrum displayed a series of strong peaks related to Ni, Co, O and C species, indicating the main chemical elements of the NiCo_2_O_4_. In Figure 2d, two states of Co^2+^ and Co^3+^ were clearly observed according to Gaussian fitting. Specifically, the fitting peaks at 779.3 eV and 794.3 eV were ascribed to Co^3+^. Another two fitting peaks at 781.0 eV and 795.8 eV were ascribed to Co^2+^ [17]. Analogously, the Ni 2p spectra included two kinds of nickel species of Ni^2+^ and Ni^3+^ in Figure 2e. The fitting peaks at 854.0 eV and 871.7 eV were ascribed to Ni^2+^, while the fitting peaks at 855.9 eV and 873.9 eV were related to Ni^3+^ [18]. As shown in Figure 2f, the fine spectrum of O 1s displayed three peaks originated from M-O-M, C-O=C and O=C. The fitting peak of M-O-M at 528.5 eV was the typical metal-oxygen bond [19]. C-O=C at a binding energy of 530.5 eV corresponded to the high number of defect sites containing low oxygen coordination [20]. O=C at a binding energy of 531.7 eV could be ascribed to the multiplicity of physisorbed water at and within the surface [17,21]. The results of XPS demonstrated a mixed valence containing Co^2+^, Co^3+^, Ni^2+^ and Ni^3+^, which was consistent with previous reports [22]. The complex electronic states of Ni^2+^/Ni^3+^ and Co^2+^/Co^3+^ could afford enough active sites for methanol oxidation, which may be one of the important factors contributing to the high electrocatalytic performance.

The surface morphologies of the CuO_2_ templates, NiO NCs and Co_3_O_4_ NCs were examined and displayed in Figure 3. As shown in Figure 3a, the Cu_2_O templates were uniformly dispersed, which was conducive to the adsorption of Ni^2+^ and Co^2+^. The surface of Cu_2_O was smooth and the edge size was about 500 nm (Figure 3b). As observed in Figure 3c, the NiO cube was composed of a large number of stacked nanoparticles, and the NiO cube had nearly the same size compared to Cu_2_O. As shown in Figure 3d, the NiO cube displayed a hollow structure with a wall thickness of about 80 nm. Similarly, the Co_3_O_4_ cube also displayed a hollow cubic feature (Figure 3e). However, the Co_3_O_4_ NCs mainly consisted of a large number of stacked nanosheets, forming a network structure (Figure 3f).

As observed in Figure 4a, the uniformly distributed Ni-Co hydroxide precursor accurately replicated the cubic structure of Cu_2_O templates and had an edge size of 500 nm. As shown in Figure 4b, the surface of Ni-Co hydroxide precursor was composed of a large number of interacted nanosheets and formed a network structure. In addition, the precursor displayed a cage-like structure and the shell thickness was about 200 nm. After calcination, the nanosheets on the surface became thicker, more compact and the thickness of the shell reduced to about 100 nm (Figure 4c). Notably, the crinkly nanosheets structure was clearly investigated in Figure 4d and the result was consistent with the SEM image. The SAED pattern in the insert of Figure 4d exhibited well-defined rings, revealing the polycrystalline nature of NiCo_2_O_4_ NCs. As observed in Figure 4e,f, the lattice spacing of 0.241 nm, 0.209 nm, 0.245 nm and 0.205 nm corresponded to the (111), (200), (311) and (400) planes of spinel NiCo_2_O_4_, respectively. The results were consistent with the XRD analysis and a previous report [22]. On the basis of the above discussion, NiCo_2_O_4_ NCs were constructed by the combination of the CEP method and post calcination. The highly porous structure provided sufficient active sites and mass transport channels, which are beneficial for electrocatalytic kinetics, leading to high electrocatalytic activity [23,24].

### 3.2. Electrocatalytic Activity of NiCo_2_O_4_ NCs/GCE towards Methanol

The electrocatalytic activity of NiCo_2_O_4_ NCs/GCE and the contrast samples was detailly evaluated by CV and EIS. Figure 5a shows the CV curves of the three electrodes in 1 M KOH in the absence of methanol. The distinct pairs of redox peaks were observed in all the three CV curves. The redox peaks of NiO NCs/GCE corresponded to the reversible transition of Ni ions, such as Ni^2+^/Ni^3+^ [25]. Similarly, the redox peaks of Co_3_O_4_ NCs/GCE were attributed to the transition between Co^2+^/Co^3+^ or Co^3+^/Co^4+^ [26]. The CV curve of NiCo_2_O_4_ NCs/GCE exhibited a much larger enclosed area than those of the Co_3_O_4_ NCs/GCE and NiO NCs/GCE. This may be due to the fact that NiCo_2_O_4_ was generally regarded as a binary TMO, which has more complicated redox electrical pairs [27]. As displayed in Figure 5b, the electrocatalytic current towards methanol on the NiCo_2_O_4_ NCs/GCE, Co_3_O_4_ NCs/GCE and NiO NCs/GCE can be clearly observed compared to Figure 5a, demonstrating that all the three electrodes showed catalytic activity towards methanol. Notably, the NiCo_2_O_4_ NCs/GCE presented a larger catalytic current than the other two electrodes. With the potential rising to 0.45 V, the current of NiCo_2_O_4_ NCs/GCE was 3.16 and 9.11 times that of Co_3_O_4_ NCs/GCE and NiO NCs/GCE, respectively. In addition, the onset potential towards methanol oxidation on the NiCo_2_O_4_ NCs/GCE was about 0.29 V (Appendix A, which was lower than those of Co_3_O_4_ NCs/GCE (0.34 V, Appendix A) and NiO NCs/GCE (0.38 V, Appendix A), revealing higher electrocatalytic activity. As shown in Figure 5c, the EIS was carried out in 1 M KOH containing 0.5 M methanol and the equivalent circuit is displayed in the insert. In the circuit, *R_s_*, *C*, *R_ct_* and *Z_w_* were the internal resistance, redox capacitance, charge transfer resistance and Warburg resistance, respectively [28,29]. Notably, the *R_ct_* value of NiCo_2_O_4_ NCs/GCE (4.4 KΩ) was obviously lower than those of Co_3_O_4_ NCs/GCE (10.8 KΩ and NiO NCs/GCE (18.3 KΩ), indicating fast electron transfer rate within the electrode or at the electrode/electrolyte interface. The lower charge transfer resistance was related to the anisotropic feature of building blocks and relatively high conductivity of NiCo_2_O_4_. At low frequencies, the NiCo_2_O_4_ NCs/GCE displayed larger *Z_w_* than Co_3_O_4_ NCs/GCE and NiO NCs/GCE, revealing lower ion diffusion resistance. The lower ion diffusion resistance might be attributed to ample diffusion channels afforded by the interacted NiCo_2_O_4_ nanosheets. In order to support the kinetics analysis of EIS, the surfaces area and porosity of NiCo_2_O_4_ NCs were tested by BET. In Figure 5d, the curve presents a H_3_-type hysteric loop in the range of 0.45–1.0, indicating a typical mesoporous characteristic [30,31]. The mean pore size of NiCo_2_O_4_ NCs/GCE was around 9 nm, which was ideal for the diffusion of methanol [32]. Moreover, the specific surface area and pore volume were 38.3 m^2^ g^−1^ and 0.2 cm^3^ g^−1^, respectively, which were both higher than those of the precursor (30.0 m^2^ g^−1^, 0.1 cm^3^ g^−1^, Appendix A). The large specific surface area provided abundant active sites for methanol catalysis, and the appropriate pore volume provided ordered diffusion channels for rapid transport [33]. In short, NiCo_2_O_4_ NCs/GCE exhibited rich redox active sites and transmission channels, leading to excellent electrocatalytic activity.

The chronoamperometry is an effective tool to investigate electrochemical stability of the electrocatalyst. As shown in Figure 6a, the electrochemical stability of the NiCo_2_O_4_ NCs/GCE, Co_3_O_4_ NCs/GCE and NiO NCs/GCE for methanol oxidation at 0.45 V was investigated. Notably, the NiCo_2_O_4_ NCs/GCE displayed largest electrocatalytic current towards 0.5 M methanol. The current of NiCo_2_O_4_ NCs/GCE displayed a decrease at the initial stage due to poisoning of the intermediates, and then kept a relatively steady value until 1100 s [34,35]. The final current still maintained 85% of its original value, which was three times of the Co_3_O_4_ NCs/GCE and thirteen times of the NiO NCs/GCE. The CV tests were carried out for 1000 cycles to further investigate the stability of NiCo_2_O_4_ NCs/GCE. The maximum current density presented an 8% decrease at the 500th cycle, and maintained 80% of the initial value after 1000 cycles. The hierarchical porous structure provided sufficient interspaces for accommodation of volume change and structural strain during electrocatalysis, resulting in excellent long-term stability towards methanol. 

## 4. Conclusions

In summary, the NiCo_2_O_4_ NCs were successfully synthesized through the CEP method combined with a post-annealing process. The designed NiCo_2_O_4_ NCs were constructed through the interaction between NiCo_2_O_4_ NSs and formed a hierarchical cage-like structure. As a catalytic electrode for methanol oxidation, the NiCo_2_O_4_ NCs/GCE exhibited high electrocatalytic activity in terms of low onset potential (0.29 V) and excellent long-term stability (80% after 1000 cycles). It is demonstrated that the NiCo_2_O_4_ NCs/GCE was an ideal electrode for DMFCs and the design of hollow hierarchical structure was an effective method to obtain highly active 2D electrocatalysts.

## Figures and Tables

**Figure 1 nanomaterials-11-02667-f001:**
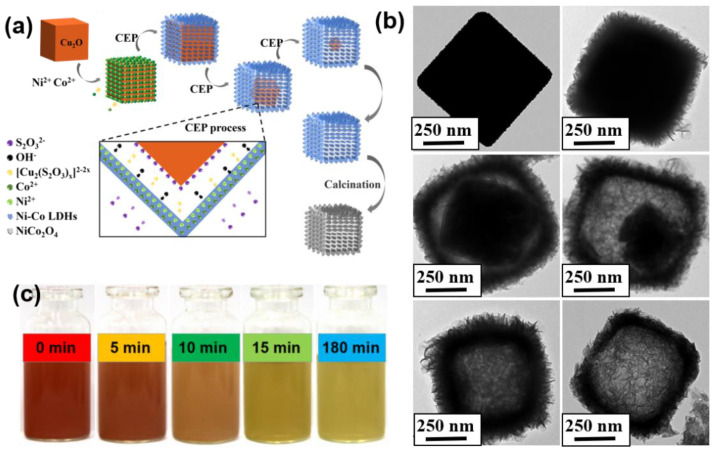
(**a**) Schematic illustration of the proposed growth mechanism of NiCo_2_O_4_ NCs. (**b**) TEM images of the products monitored at different reaction times. (**c**) Optical photographs of the suspension at different reaction time after addition of etchant.

**Figure 2 nanomaterials-11-02667-f002:**
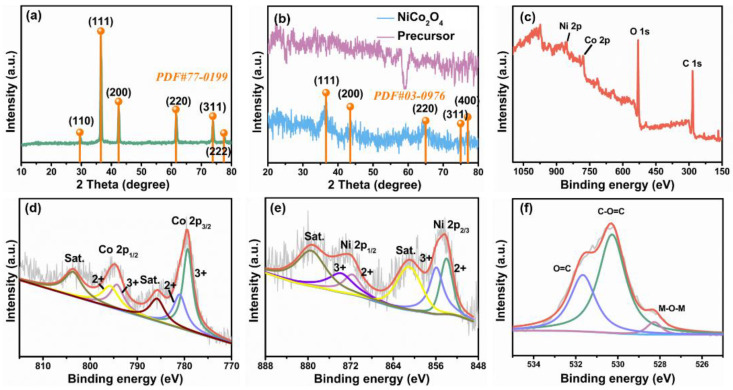
XRD pattern of (**a**) Cu_2_O, (**b**) Ni-Co hydroxide precursor and NiCo_2_O_4_; (**c**) XPS survey of NiCo_2_O_4_, (**d**) Co 2p, (**e**) Ni 2p and (**f**) O 1s.

**Figure 3 nanomaterials-11-02667-f003:**
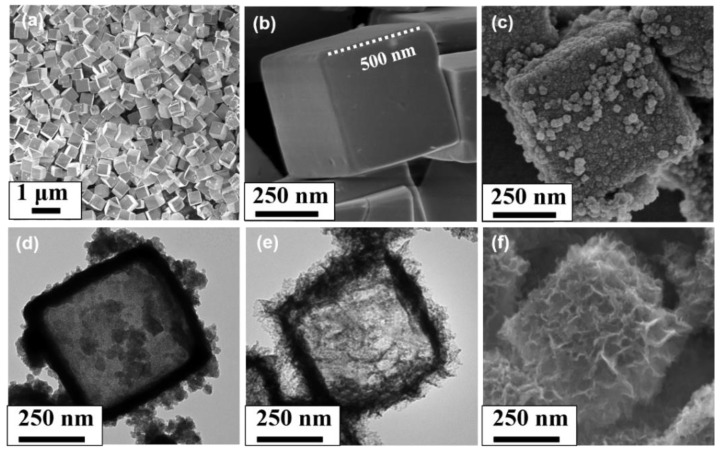
SEM images of (**a**,**b**) Cu_2_O and (**c**) NiO NCs; TEM images of prepared (**d**) NiO NCs and (**e**) Co_3_O_4_ NCs; (**f**) SEM image of prepared Co_3_O_4_ NCs.

**Figure 4 nanomaterials-11-02667-f004:**
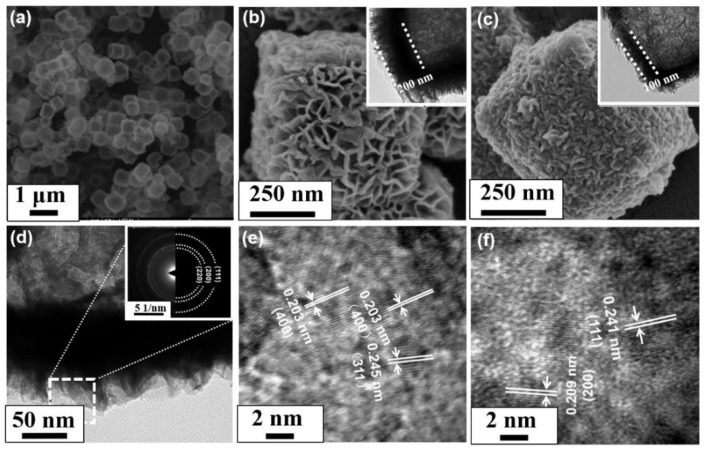
(**a**) TEM images of Ni-Co hydroxide precursor and (**b**,**c**) SEM of prepared Ni-Co hydroxide precursor and NiCo_2_O_4_ NCs; (**d**–**f**) HRTEM of prepared NiCo_2_O_4_ NCs the inset shows SAED.

**Figure 5 nanomaterials-11-02667-f005:**
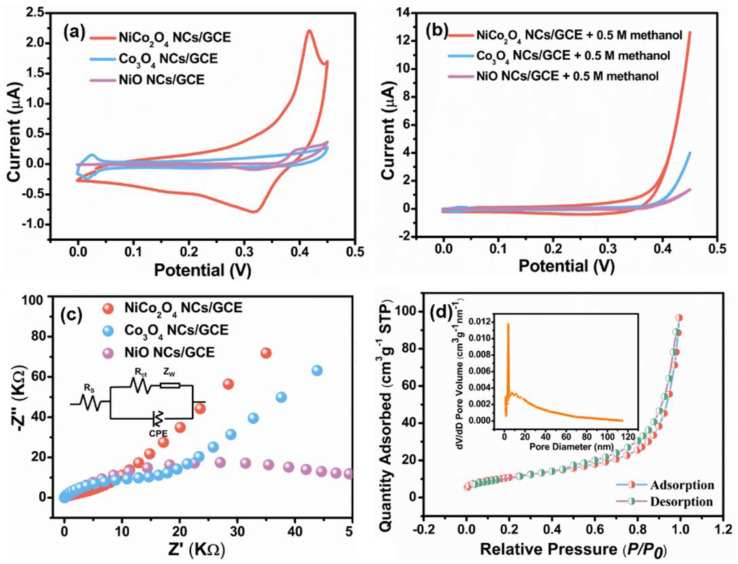
CV curves of NiCo_2_O_4_ NCs/GCE, Co_3_O_4_ NCs/GCE and NiO NCs/GCE in 1 M KOH (**a**) without methanol and (**b**) with 0.5 M methanol at 50 mV s^−1^; (**c**) EIS plots of NiCo_2_O_4_ NCs/GCE, Co_3_O_4_ NCs/GCE and NiO NCs/GCE in 1 M KOH with 0.5 M methanol; (**d**) N_2_ adsorption–desorption isotherms of the NiCo_2_O_4_ NCs/GCE.

**Figure 6 nanomaterials-11-02667-f006:**
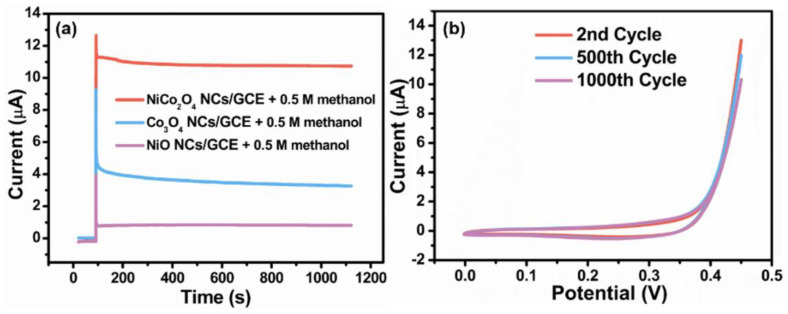
(**a**) The Chronoamperometry curves of NiCo_2_O_4_ NCs/GCE, Co_3_O_4_ NCs/GCE and NiO NCs/GCE in 0.5 M methanol. (**b**) The CV stability test of NiCo_2_O_4_ NCs/GCE in 0.5 M methanol at a scanning rate of 50 mV s^−1^.

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
