# Peer review of "Design of Hierarchical NiCo2O4 Nanocages with Excellent Electrocatalytic Dynamic for Enhanced Methanol Oxidation"

_nanomaterials, 2021, doi:10.3390/nano11102667_

Round 1

Reviewer 1 Report

The authors describe a very interesting work on electrochemical nanomaterials based on NiCo2O4 nanocubes showing good performance and cycling stability for direct methanol fuel cells.

here are some elements can be improved to inprove the impact of the work:

(1) I suggest to avoid acronyms in the abstract and to put it in present tense while keeping past tense to previous results from the literature. Moreover, the “gap” which is filled by this work should be expressed in the abstract: what problem are you facing with this strategy?

(2) The statement on the promising results in the literature on nano cages is not supported by appropriate references (see line 50-65, page2)

(3) Please avoid repeating the same quantitative results in abstract, end of introduction, results and conclusions! Abstract should only give general and main result, the introduction should state the strategy and the overall meaning of the results in the context and in the conclusions the results should be put in the larger context to drive a general conclusion.

(4) Please comment on the term nanocages, nano could be justified by the cage thickness below 100 nm, however the cage dimensions is above 500 nm, i.e. submicron. 

(5) I have difficulty in understanding the caption of Fig.3, please comment the image of each panel separately including large and small features as seen in panel c.

(6) The XPS fit of the O1s peak shows a surprisingly large O-M component, please explain. I suggest to take a look to a recent work on O1s fitting (https://doi.org/10.1021/acsami.0c08297) and try to lower the peak width to avoid overlap and underestimation of other chemical components.

Reviewer 2 Report

This is a thorough piece of work describing the preparation and characterization of NiCoO4 mixed oxides with a nano-cage structure. Compared to the pure component oxides, NiO and CoO, these mixed oxide structures show a largely enhanced activity as methanol oxidation electro-catalysts. The paper is clearly structured and well written; the methods used are very appropriate, and the results and conclusions are fully convincing.

There are only two minor points the authors should clarify before publication:

Page 4, line 161: The presence of physicochemically (physisorbed?) water after calcination appears rather unlikely.

Page 4 and Fig. 2: The authors should definitely include the high-resolved XPS region of the Cu2p emission (>900 eV) in order to verify the “complete disappearance of Cu20” as claimed on page 3, line 127).

After clarification of these two issues the paper should be published.

Round 2

Reviewer 1 Report

The manuscript looks to me now suitable for acceptance in its present form.

regards